# The Catalytic Degradation of the Inflammatory Drug Diclofenac Sodium in Water by $Fe^{2+}$/Persulfate, $Fe^{2+}$/Peroxymonosulfate and $Fe^{2+}$/$H_2O_2$ Processes: A Comparative Analysis

**Faiza Rehman** [1,*]**, Waqas Ahmad** [1]**, Nazish Parveen** [1]**, Syed Khuram Zakir** [1]**, Sanaullah Khan** [2,3] **and Changseok Han** [4,5,*]

1 Department of Chemistry, University of Poonch, Rawalakot 12350, Pakistan
2 Department of Chemistry, Women University Swabi, Swabi 23430, Pakistan
3 Department of Biochemistry, Women University Swabi, Swabi 23430, Pakistan
4 Department of Environmental Engineering, INHA University, Incheon 22212, Republic of Korea
5 Program in Environmental & Polymer Engineering, Graduate School, INHA University, Incheon 22212, Republic of Korea
* Correspondence: faizaawanchem@yahoo.com (F.R.); hanck@inha.ac.kr (C.H.)

**Abstract:** Diclofenac sodium was extensively used for treating arthritis, osteoarthritis and skeletal muscular injuries, which ultimately caused troubles for aquatic organisms as well as human beings. In this study, homogeneous catalytic advanced oxidation processes, including $Fe^{2+}$/persulfate, $Fe^{2+}$/peroxymonosulfate and $Fe^{2+}$/$H_2O_2$, were used for the degradation of diclofenac sodium in water, without using UV-C light. About 89, 82 and 54% DCF sodium was decomposed by $Fe^{2+}$/persulfate, $Fe^{2+}$/peroxymonosulfate and $Fe^{2+}$/$H_2O_2$, respectively, in 60 min. The degradation of diclofenac sodium followed the *pseudo first-order* kinetics, in all cases. The degradation efficiency of diclofenac sodium was significantly affected in the presence of various anions, such as $NO_3^-$, $HCO_3^-$ and $SO_4^{2-}$. The mineralization studies revealed 62, 45 and 32% total carbon removal by $Fe^{2+}$/persulfate, $Fe^{2+}$/peroxymonosulfate and $Fe^{2+}$/$H_2O_2$, respectively, in 60 min. In addition, the degradation byproducts of diclofenac sodium were determined by FTIR analysis. The results revealed that the $Fe^{2+}$/oxidant system, particularly $Fe^{2+}$/persulfate, was a promising technology for the elimination of toxic pharmaceuticals, such as diclofenac sodium, from the water environment.

**Keywords:** diclofenac sodium; homogeneous catalysis; advanced oxidation processes; water treatment

## 1. Introduction

Pharmaceuticals are medicinal compounds that are extensively used for treating human and veterinary diseases all over the world [1]. Pharmaceutical compounds are being used for several other purposes, including cosmetics, food supplements, metabolites and their intermediate products. In many cases, they may become toxic contaminants for different species in the environment [2,3]. The remains of used pharmaceuticals are frequently detected in sewage and wastewater systems [4]. Trace concentrations of pharmaceuticals are reported to cause water pollution [5]. An increasing level of toxic organic compounds seeps down into lakes, rivers and streams, and are hazardous to aquatic life and the environment [6–8].

Antibiotics and antidepressant pharmaceutical compounds are not completely metabolized, and fractions of those compounds are excreted out from the bodies of humans and animals through urine and feces, which are treated in sewage treatment plants (STPs) [9]. However, these pharmaceuticals are not fully removed in STPs by conventional physical and biological treatment processes [10] and may require further chemical treatment before being discharged into the environment.

Diclofenac sodium (DCF sodium, i.e., $C_{14}H_{10}Cl_2NO_2Na$) is a commonly used anti-inflammatory drug and electroactive painkiller, heavily utilized in human beings and animals [11]. The chemical structure of DCF sodium is given in Figure 1. DCF sodium is used as an analgesic, antirheumatic and antiarthritic drug [12]. It may relieve pain related to neuralgia, cancer, post-traumatic, post-operative and soft tissue complaints, and is used in the treatment of other inflammatory diseases [13]. The potassium and sodium salts of DCF soluble in water were used for oral administration. DCF sodium is heavily produced, and 490 tons of the drug are consumed annually all over the world [14,15]. Although the acute toxicity of DCF sodium is low, i.e., $EC_{50}$ value of 33.26 mg $L^{-1}$ (estimated by EPI Suite Model (EPA 2008)), its long-term effect on living organisms is highly adverse [16]. A high consumption of DCF sodium can cause liver and kidney problems, increase uric acid concentration in body and cause gout, which can lead to death [17]. The long-term exposure of DCF sodium to the environment is toxic to the health of fish, causing renal lesions and alteration of gills [18]. A DCF sodium concentration of 5 µg/L is suggested as the lowest observed concentration in water, which may cause renal lesions to aquatic organisms [19]. DCF sodium was also detected in urban wastewater treatment plants (UWWTPs), in concentrations ranging from 2–10 µg/L [20]. Owing to its heavy use, DCF sodium was ubiquitously reported in various environmental compartments [21]. DCF sodium is quite stable and non-biodegradable in the environment, and conventional WWTPs are inefficient for the removal of DCF sodium from water [22]. Due to the vast application and high toxicity, there is a dire need to develop effective methods for removing DCF sodium from the water environment.

**Figure 1.** Sodium (2[(2,6-diclorophenyl)amino]phenyl acetate).

Advanced oxidation processes (AOPs), such as the Fenton process, photocatalysis, radiolysis, and electrocatalysis, are promising technologies for the removal of recalcitrant organic pollutants from water [23–26]. The AOPs are characterized by the generation of highly oxidizing species, capable of oxidative degradation of organic compounds [27,28], leading to mineralization into non-toxic or less toxic compounds and more biodegradable compounds [29,30]. The hydroxyl radical ($^{\bullet}OH$) and sulfate radical ($SO_4^{\bullet-}$) based AOPs are growing technologies for the degradation of organic pollutants in water, and hydrogen peroxide ($H_2O_2$) as well as persulfate (PS) or peroxymonosulfate (PMS) are the major precursors of these radicals. Several AOPs, including photocatalysis [31,32], sonolysis [33], sonophotocatalysis [34], UV/PS/$Fe^{2+}$, UV/PMS/$Fe^{2+}$ or UV/$H_2O_2$/$Fe^{2+}$ [15], UV/$H_2O_2$ [35], UV/$O_3$/PS [36] and gamma irradiation [37], are used for the degradation of DCF sodium in water. Most of the reported AOPs are based on ultrasonic or UV light for the activation of the oxidants [15,33–35]. Both homogeneous and heterogeneous photocatalyses were employed for the degradation of DCF sodium in water [38]. Many different kinds of photocatalysts, including $TiO_2$ [39,40], $RuTe_2$/black $TiO_2$ [41], Ag modified g-$C_3N_4$ composites [42], ZnO-$WO_3$ [43], ZnO [44] and F-doped ZnO [45], were employed for the degradation of DCF sodium in water. $Fe^{2+}$ is a widely used homogeneous and heterogeneous catalyst/photocatalyst for the degradation of a large number of recalcitrant organic pollutants in water [46,47]. Iron ($Fe^{2+}$) is a comparatively environmentally friendly element [48], and traces of iron can be found in surface and ground water resources [49]. The $Fe^{2+}$ found in natural water resources may be involved in the activation of oxidants, causing the oxidation (or degradation) of organic pollutants in water [49]. Thus, Fenton, photo-Fenton and electro-Fenton processes were efficiently used for the removal of organic pollutants, especially pharmaceutical compounds, from water [50–52]. $Fe^{2+}$ is a promising

alternative for the activation of oxidants such as $H_2O_2$, PS or PMS; however, studies on the degradation of DCF sodium by the $Fe^{2+}$/oxidant system are still very limited. One study shows the presence of various inorganic ions, especially $NO_3^-$, $HCO_3^-$ and $SO_4^{2-}$, in natural or ground water resources [53,54]. The efficiency of the AOPs may be affected by the water quality parameters, including inorganic anions (i.e., $NO_3^-$, $HCO_3^-$ and $SO_4^{2-}$), which is a subject of interest for practical applications purposes.

In this study, homogeneous catalytic processes, including $Fe^{2+}$/PS, $Fe^{2+}$/PMS and $Fe^{2+}$/$H_2O_2$ processes, without using UV light, are used for the degradation of DCF sodium in water. The effect of process parameters, such as the initial concentrations of DCF sodium, $Fe^{2+}$ and oxidants (i.e., $H_2O_2$, PS and PMS), are investigated. The effect of various inorganic anions, i.e., $NO_3^-$, $HCO_3^-$ and $SO_4^{2-}$ on the degradation efficiency of DCF sodium are investigated. The degradation byproducts of DCF sodium are studied with FTIR analysis. Additionally, the mineralization of DCF sodium by $Fe^{2+}$/$H_2O_2$, $Fe^{2+}$/PS and $Fe^{2+}$/PMS systems is measured by the total carbon (TC) content removal. The results reveal that the $Fe^{2+}$/oxidant system, particularly $Fe^{2+}$/PS, is a promising technology for the elimination of toxic pharmaceuticals, such as DCF sodium, from the water environment. Information concerning the degradation of an anti-inflammatory drug, DCF sodium, by the $Fe^{2+}$/oxidant system is very new, and the results of this study could be effectively used to protect the health of humans, animals and the ecosystem.

## 2. Materials and Methods

### 2.1. Materials

DCF Sodium ($\geq$98%) characterized by a water solubility of 237 mg $L^{-1}$ at 25 °C, potassium persulfate (PS) and peroxymonosulfate (PMS) ($2KHSO_5 \cdot KHSO_4 \cdot K_2SO_4$) were purchased from Sigma-Aldrich. Hydrogen peroxide ($H_2O_2$, 50%, *v/v*), $FeSO_4 \cdot 7H_2O$, $CoCl_2$, $AlCl_3$, EDTA, $CuSO_4$, $(NH_3)HCO_3$, $Co(NO_3)_2$, KCl, $CH_3COONa$ and $CaCl_2$ were of an analytical grade and were purchased from Fisher Scientific. They were used without any further treatment for all experiments.

### 2.2. Degradation Experiment

The degradation of DCF sodium ($C_0$ = 0.3 mM) was carried out in a bench scale reactor with a volume of 50 mL, containing homogeneous solutions of $Fe^{2+}$ ($C_0$ = 0.5 mM) and PS, PMS or $H_2O_2$ ($C_0$ = 10 mM). The efficiency of the Fenton-like processes is considered high at an acidic pH since $Fe^{2+}$ might be precipitated as $Fe^{3+}$ at a neutral or basic pH. Hence, the degradation experiments were carried out at a pH of 4. The experiments were performed in triplicate unless stated otherwise. The error bars in the figures denote the standard error of the mean. All the solutions were prepared in Milli-Q water (Resistivity 18.2 M$\Omega$ cm).

### 2.3. Analytical Methods

The DCF sodium was analyzed by using a UV-Visible spectrophotometer (SPECORD 210 PLUS) at a wavelength of 276 nm. The FTIR (Alpha FTIR Spectrometer) analysis was used for the detection of different functional groups of the degradation byproducts of DCF sodium. The samples in the liquid state were used for the FTIR analysis. A drop of liquid sample was used for scanning. The peaks obtained from the degradation byproducts were also matched with the literature to identify the classes of compounds. For the analysis of TC removal, a Shimadzu TOC-LCSH/CSN TOC analyzer was used.

## 3. Results and Discussion

### 3.1. Degradation of DCF Sodium by $Fe^{2+}$/PS, $Fe^{2+}$/PMS and $Fe^{2+}$/$H_2O_2$ Systems

The degradation of DCF sodium in water by $Fe^{2+}$/PS, $Fe^{2+}$/PMS and $Fe^{2+}$/$H_2O_2$ systems is shown in Figure 2. The results show that 89, 82 and 54% degradation of DCF sodium was achieved by the $Fe^{2+}$/PS, $Fe^{2+}$/PMS and $Fe^{2+}$/$H_2O_2$ systems, respectively, in 60 min. Additionally, the degradation of DCF sodium by $Fe^{2+}$, PS, PMS or $H_2O_2$ alone was less than 5% during 60 min (results not shown in the Figure). The degradation of

DCF sodium was attributed to the generation of a sulfate radical anion ($SO_4^{\bullet-}$), hydroxyl radical ($^\bullet OH$) or both $SO_4^{\bullet-}$ and $^\bullet OH$, by the $Fe^{2+}$/PS, $Fe^{2+}$/$H_2O_2$ or $Fe^{2+}$/PMS systems, respectively, generated via reactions (1)-(3) [55,56]. The degradation of DCF sodium in water by $SO_4^{\bullet-}$ and/or $^\bullet OH$ is reported in the literature elsewhere. Jabbari et al. reported an 89% degradation of diclofenac in water by the $O_3$/UV/$S_2O_8$ system after 30 min, attributed to the reaction of $SO_4^{\bullet-}$ [36]. Yu et al. reported a 60% degradation of diclofenac by gamma irradiation at a 20 k Gy radiation dose, attributed to the reaction of $^\bullet OH$ [37]. Pourzamani et al. found 78% degradation of DCF in 90 min via $^\bullet OH$ oxidation using a graphite electrochemical reaction [57]. In our previous study, a 98% degradation of diclofenac sodium was achieved by UV/PMS/$Fe^{2+}$ in 60 min [15]. Tian et al. [58] found the mineralization efficiency of the antibiotics by $Fe^{2+}$/PDS was higher than $Fe^{2+}$/$H_2O_2$, even though the highest mineralization efficiency was shown by the $Fe^{2+}$/PMS system. Wang and Wang [59] reported the degradation efficiency of sulfamethoxazole by $Fe^{2+}$/PS was similar to the $Fe^{2+}$/$H_2O_2$ system. On the other hand, Song et al. [55] found the degradation efficiency of the flame retardant triphenyl phosphate by $Fe^{2+}$/$H_2O_2$ was higher than the $Fe^{2+}$/PS system, attributed to the higher radical intensity of $Fe^{2+}$/$H_2O_2$ compared to $Fe^{2+}$/PS. In another study by Wang and Wang [60], it was also shown that the degradation efficiency of trimethoprim by the Fenton process was higher than the $Fe^{2+}$/persulfate process. However, Wang and Wang [60] found that in the case of actual wastewater samples, the removal efficiency of trimethoprim by the Fenton process was lower than the $Fe^{2+}$/persulfate system.

$$Fe^{2+} + H_2O_2 \rightarrow Fe^{3+} + {}^\bullet OH + OH^- \tag{1}$$

$$Fe^{2+} + S_2O_8{}^{2-} \rightarrow Fe^{3+} + SO_4{}^{\bullet-} + SO_4{}^{2-} \tag{2}$$

$$Fe^{2+} + HSO_5{}^- \rightarrow Fe^{3+} + SO_4{}^{\bullet-} + OH^- \tag{3}$$

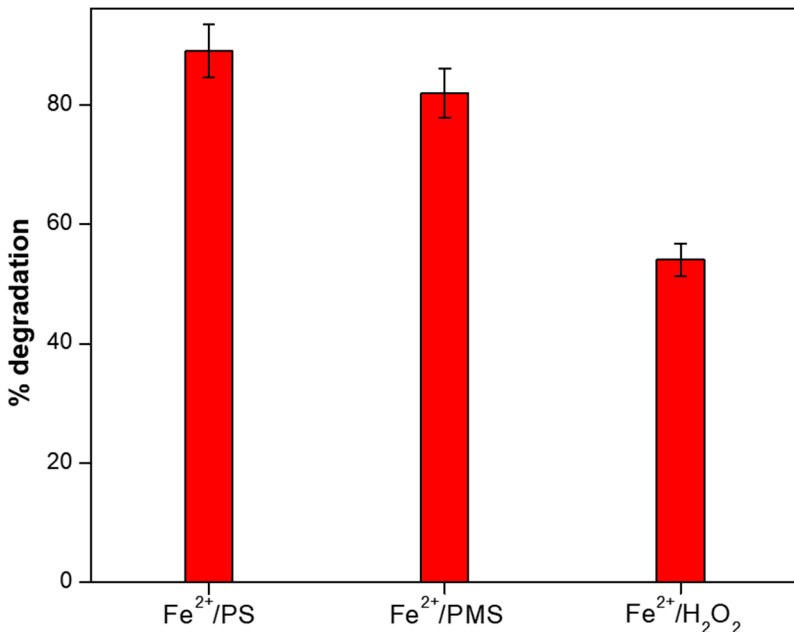

**Figure 2.** Degradation efficiency of DCF sodium by $Fe^{2+}$/$H_2O_2$, $Fe^{2+}$/PS and $Fe^{2+}$/PMS systems in 60 min. Reaction conditions: [DCF sodium]$_0$ = 0.3 mM, [$Fe^{2+}$]$_0$ = 0.5 mM, [$H_2O_2$]$_0$ = [PS]$_0$ = [PMS]$_0$ = 10 mM, pH = 4.

### 3.2. The effect of the Initial Concentration of DCF Sodium

The degradation of DCF sodium by $Fe^{2+}$/oxidant processes, including $Fe^{2+}$/PS, $Fe^{2+}$/PMS and $Fe^{2+}$/$H_2O_2$ systems, was carried out using different initial concentrations

of DCF sodium, i.e., 0.1, 0.5 and 1.0 mM, and the results are shown in Figure 3. The degradation efficiency of DCF sodium by $Fe^{2+}$/PS was 94, 89 and 80% when the initial concentration of DCF sodium was 0.1, 0.5 and 1.0 mM, respectively. Similarly, the degradation efficiency of the $Fe^{2+}$/PMS system was 90, 82 and 74% when the initial concentration of DCF sodium was 0.1, 0.5 and 1.0 mM, respectively. Meanwhile, the degradation efficiency of the $Fe^{2+}$/$H_2O_2$ system was 65, 54 and 40% at 0.1, 0.5 and 1.0 mM initial concentrations of DCF sodium, respectively. The results show that the degradation efficiency of DCF sodium by the $Fe^{2+}$/oxidant processes was decreased with the increasing concentrations of DCF sodium. The increased competition between the reaction byproducts and the parent compound for the reactive species (i.e., $SO_4^{\bullet-}$ and $^{\bullet}OH$) could be mainly responsible for the reduced degradation efficiency at high initial concentrations of the pollutant [61]. This result was consistent with our previous paper, indicating the degradation efficiency of lindane by photo-Fenton-like processes decreased with the increase in the initial concentration of the pollutant [56]. Furthermore, the results showed that the plots of $\ln(C/C_0)$ vs. time were straight lines (i.e., $R^2 > 0.95$) in all cases, indicative of the *pseudo-first-order* kinetics with respect to the concentration of the pollutant, i.e., DCF sodium (Figure 4). Zhang et al. [62] showed the degradation of Norfloxacin in water by nanoscale zero-valent iron-activated persulfate (nZVI/PS) process followed the pseudo-first-order kinetics consistent with our results.

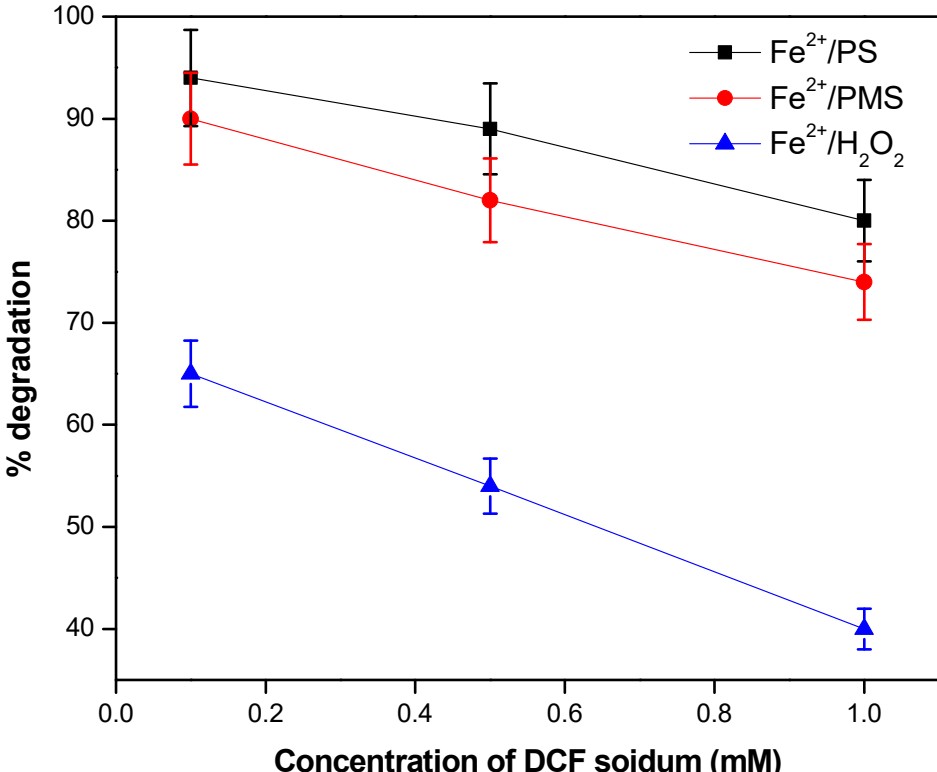

**Figure 3.** The effect of the initial concentration of DCF sodium on the degradation efficiency of DCF by the $Fe^{2+}$/$H_2O_2$, $Fe^{2+}$/PS and $Fe^{2+}$/PMS systems. Reaction conditions: [DCF sodium]$_0$ = 0.1–1.0 mM, [$Fe^{2+}$]$_0$ = 0.5 mM, [$H_2O_2$]$_0$ = [PS]$_0$ = [PMS]$_0$ = 10 mM, pH = 4.

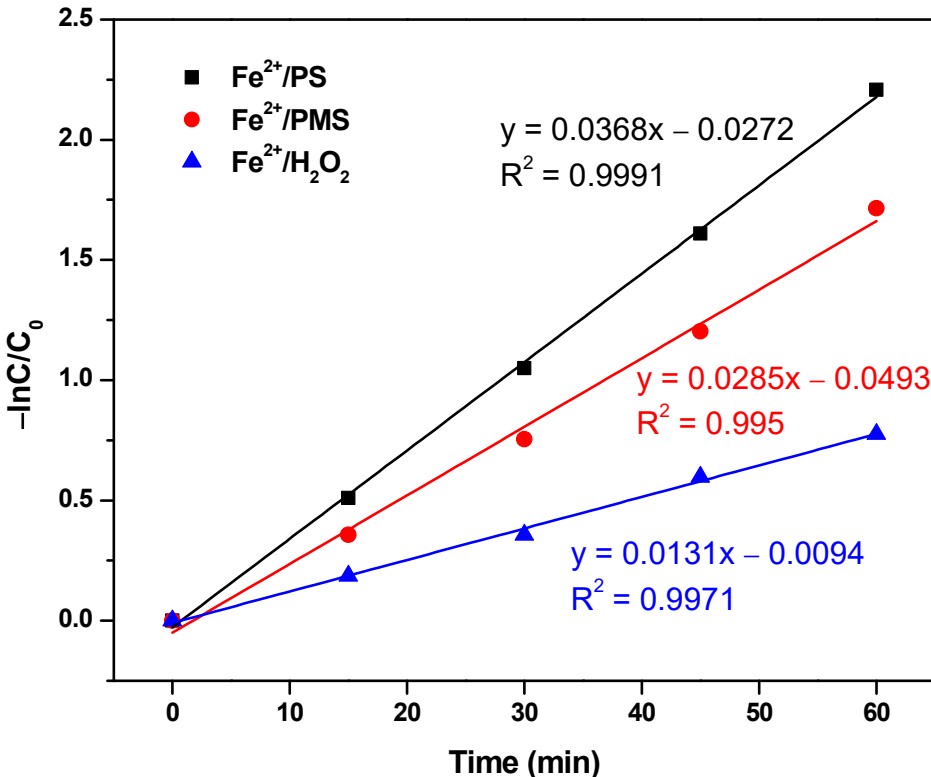

**Figure 4.** Pseudo-first-order kinetics for degradation of DCF sodium by $H_2O_2/Fe^{2+}$, $PS/Fe^{2+}$·and $PMS/Fe^{2+}$ reactions.

### 3.3. The Effect of Concentrations of $H_2O_2$, PS and PMS

Different initial concentrations of $H_2O_2$—i.e., 5, 10 and 20 mM—were used to study its effect on the degradation of DCF sodium by the $Fe^{2+}/H_2O_2$ system, and the results are shown in Figure 5. From Figure 5, it is clear that when the concentration of $H_2O_2$ was increased from 5 to 20 mM, the degradation of DCF sodium was also increased from 47 to 57% after 60 min. This result was attributed to the increased concentration of $^\bullet OH$ with the increasing concentration of $H_2O_2$. The degradation efficiency of DCF sodium by the $Fe^{2+}/PS$ system increased from 76 to 96% in 60 min when the concentration of PS was increased from 5 to 20 mM (Figure 5). This result is explained by the high concentration of $SO_4^{\bullet-}$ produced at the increased concentration of PS.

By increasing the concentrations of PMS from 5 to 20 mM, the degradation efficiency of DCF sodium by the $Fe^{2+}/PMS$ system was increased from 74 to 88% in 60 min, indicating the degradation efficiency of DCF sodium increased after increasing the concentration of PMS. These results were the same as explained in the literature elsewhere [63]. Using the above results, 10 mM was chosen as the optimum oxidant concentration during the degradation of DCF sodium by $Fe^{2+}$/oxidant systems.

### 3.4. The Effect of Concentrations of $Fe^{2+}$

The effect of the initial concentration of $Fe^{2+}$ on the degradation efficiency of DCF sodium by the $Fe^{2+}/H_2O_2$ system is shown in Figure 6. The results show that the degradation efficiency of DCF sodium was enhanced from 42 to 65% in 60 min when the concentration of $Fe^{2+}$ was increased from 0.1 to 1.0 mM. The higher degradation efficiency of DCF sodium at the increased concentration of $Fe^{2+}$ was attributed to an increased activation of $H_2O_2$, followed by the high concentration of $^\bullet OH$ under such conditions.

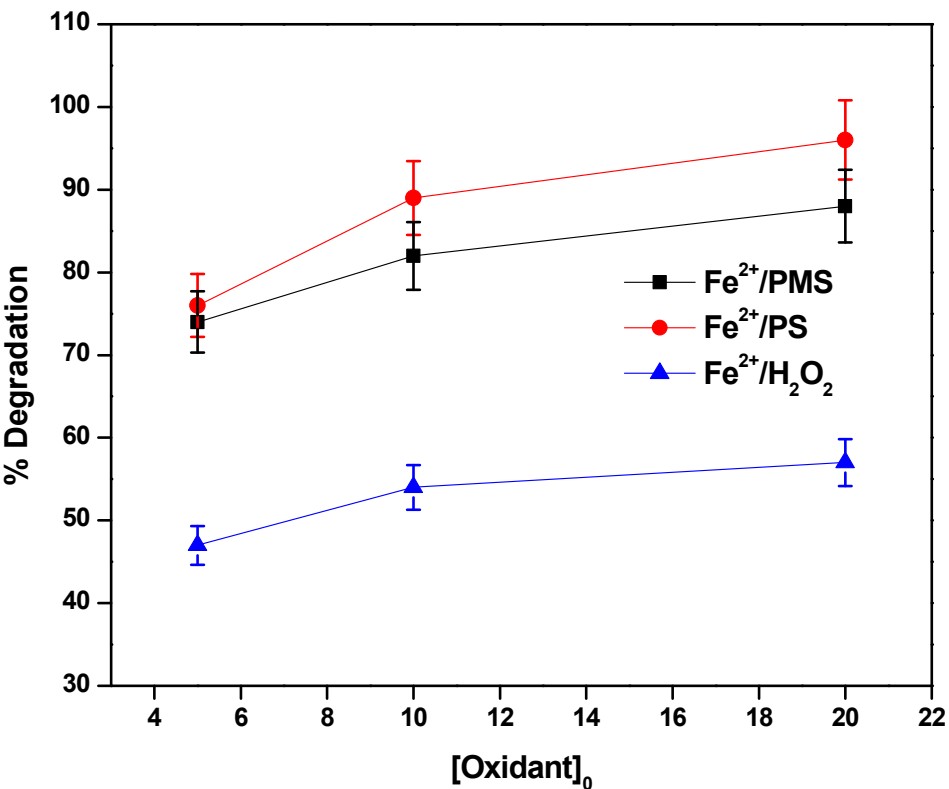

**Figure 5.** Effect of initial concentration of oxidants on the degradation efficiency of DCF sodium by the $Fe^{2+}/H_2O_2$, $Fe^{2+}/PS$ and $Fe^{2+}/PMS$ systems. Reaction conditions: $[DCF\ sodium]_0 = 0.5$ mM, $[Fe^{2+}]_0 = 0.5$ mM, $[H_2O_2]_0 = [PS]_0 = [PMS]_0 = 5$–$20$ mM, pH = 4.

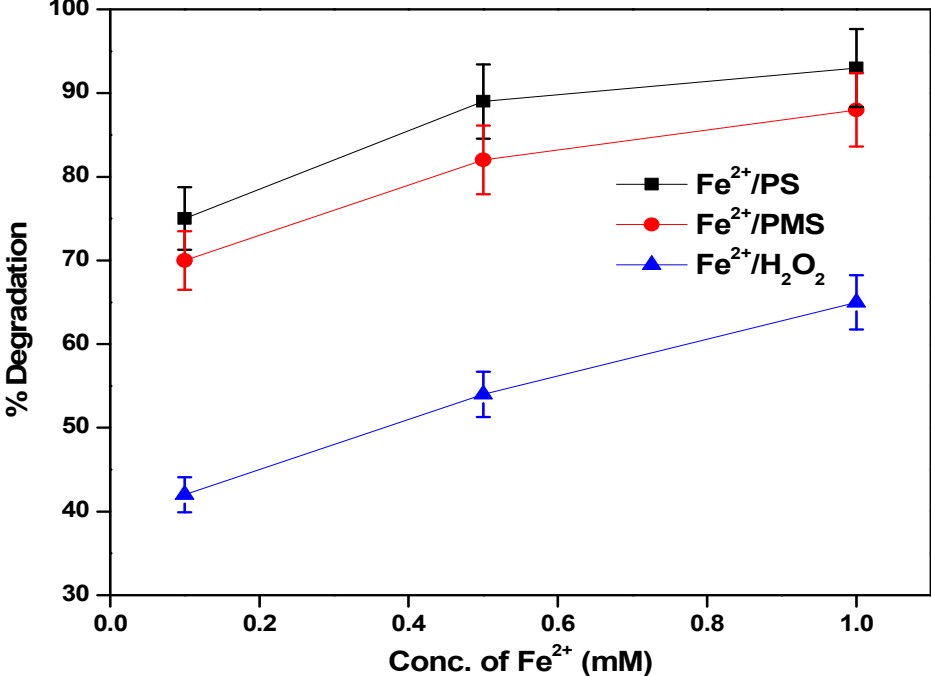

**Figure 6.** Effect of the initial concentration of $Fe^{2+}$ on the degradation of DCF sodium by the $Fe^{2+}/H_2O_2$, $Fe^{2+}/PS$ and $Fe^{2+}/PMS$ systems. Reaction conditions: $[DCF\ sodium]_0 = 0.5$ mM, $[Fe^{2+}]_0 = 0.1$–$1.0$ mM, $[H_2O_2]_0 = [PS]_0 = [PMS]_0 = 10$ mM, pH = 4.

When the concentration of $Fe^{2+}$ was increased from 0.1 to 1.0 mM using the $Fe^{2+}/PMS$ system, the degradation efficiency of DCF sodium was enhanced from 70 to 88% in 60 min,

as shown in Figure 6, attributed to the high concentration of $SO_4^{\bullet-}$ and $^{\bullet}OH$. The degradation efficiency of DCF sodium by $Fe^{2+}/PS$ was enhanced from 75 to 93% when the concentration of $Fe^{2+}$ was increased from 0.1 to 1.0 mM, attributed to the high concentration of $SO_4^{\bullet-}$, as well (Figure 6).

### 3.5. The Effect of Inorganic Anions

Natural waters may contain different inorganic ions, which can affect the degradation efficiency of the pollutants [64]. The degradation efficiency of DCF sodium by the $Fe^{2+}/oxidants$ system was carried out in the presence of some of the most commonly found inorganic anions in water, i.e., $NO_3^{1-}$, $HCO_3^{1-}$ and $SO_4^{2-}$, and the results are shown in Figure 7. It is clear from the Figure that the degradation efficiency of DCF sodium by the $Fe^{2+}/H_2O_2$ system was 50, 52 and 32% in the presence of 10 mM of $HCO_3^-$, $NO_3^-$ and $SO_4^{2-}$, respectively. Similarly, the degradation efficiency of DCF sodium by the $Fe^{2+}/PS$ or $Fe^{2+}/PMS$ systems was 83, 76 and 50% or 80, 75 and 52%, respectively. The results show that the presence of inorganic ions has a large effect on the efficiency of the $Fe^{2+}/H_2O_2$ system compared to the $Fe^{2+}/PS$ or $Fe^{2+}/PMS$ systems. A possible reason could be the relatively high reactivity of the inorganic ions towards $^{\bullet}OH$ rather than $SO_4^{\bullet-}$ generated by $Fe^{2+}/H_2O_2$ and $Fe^{2+}/PS$ systems, respectively [64]. Devi et al. [65] found the presence of $NO_3^{1-}$, $HCO_3^{1-}$ and $SO_4^{2-}$ has a negative effect on the degradation efficiency of di azo dye Bismarck Brown using $Fe^{2+}/H_2O_2/UV$ and $Fe^{2+}/PS/UV$ systems, attributed to scavenging of $^{\bullet}OH$ and $SO_4^{\bullet-}$ by inorganic anions, according to Equations (4)–(7) [66]. Song et al. [55] found the degradation efficiency of the flame retardant triphenyl phosphate by $Fe^{2+}/H_2O_2$ and $Fe^{2+}/PS$ was not obviously influenced by $NO_3^-$, which was significantly inhibited by $HCO_3^-$, and the inhibition was inversely related to $HCO_3^-$ concentrations.

$$SO_4^{2-} + {}^{\bullet}OH \rightarrow SO_4^{\bullet-} + OH^- \tag{4}$$

$$NO_3^- + {}^{\bullet}OH \rightarrow NO_3^{\bullet} + OH^- \tag{5}$$

$$HCO_3^- + {}^{\bullet}OH \rightarrow HCO_3^{\bullet-} + OH^- \tag{6}$$

$$HCO_3^- + SO_4^{\bullet-} \rightarrow CO_3^{\bullet-} + SO_4^{2-} + H^+ \tag{7}$$

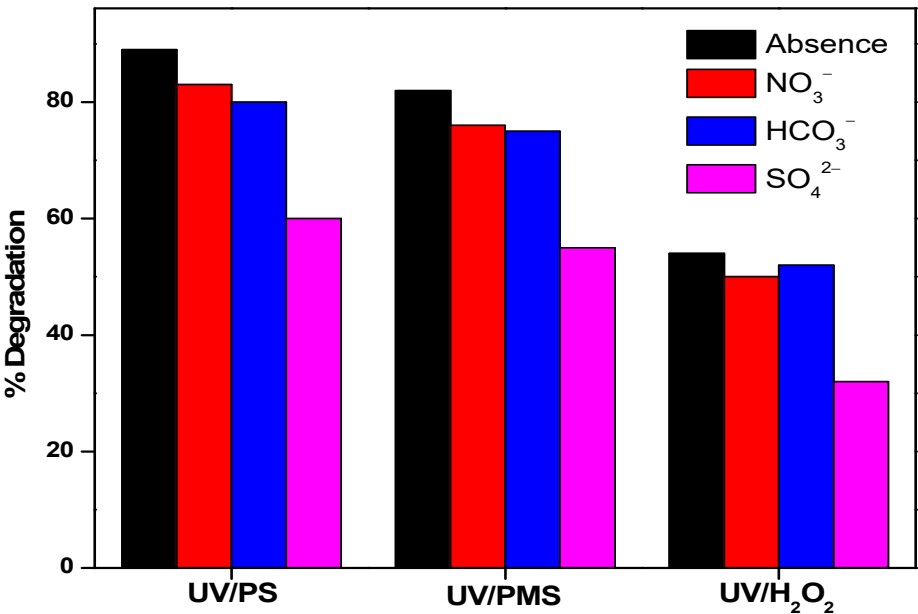

**Figure 7.** The effect of inorganic anions on the degradation efficiency of DCF sodium by using the $Fe^{2+}/H_2O_2$, $Fe^{2+}/PS$ and $Fe^{2+}/PMS$ systems. Reaction conditions: $[Anions]_0 = 10$ mM, [DCF sodium]$_0 = 0.5$ mM, $[Fe^{2+}]_0 = 0.5$ mM, $[H_2O_2]_0 = [PS]_0 = [PMS]_0 = 10$ mM, pH = 4.

### 3.6. FTIR Studies of DCF SODIUM Degradation

The functional groups of the degradation byproducts of DCF sodium by the $Fe^{2+}$/PS, $Fe^{2+}$/PMS and $Fe^{2+}$/$H_2O_2$ systems were identified by using FTIR analysis, and the results are shown in Tables 1–3, respectively. The sharp peaks of the degradation byproducts of DCF sodium were obtained at different wavenumbers. The sharp peaks at different wavenumbers revealed that DCF sodium was degraded into smaller organic and inorganic compounds, i.e., alkene, alkynes, amines, alcohols, nitrile and carbon dioxide, before the end product. The results shown in Tables 1–3 also suggested that the degradation byproducts of DCF sodium produced during the different AOPs, i.e., $Fe^{2+}$/PS, $Fe^{2+}$/PMS and $Fe^{2+}$/$H_2O_2$ systems, mostly belonged to the same group, although the number of compounds generated by $Fe^{2+}$/PMS was slightly larger than the other two systems. The production of both types of the reactive species (i.e., $SO_4^{\bullet-}$ as well as $^\bullet OH$) by the $Fe^{2+}$/PMS system might explain the generation of a large number of reaction byproducts.

**Table 1.** FTIR analysis of the functional groups of degraded byproducts of DCF sodium using the $Fe^{2+}$/PS process.

| Sr. No | Transmittance at Wavenumber (cm$^{-1}$) | Group Species | Vibration Mode | Compound Class |
|--------|------------------------------------------|---------------|----------------|----------------|
| 1 | 724, 696, 673 | C=C | Bending | Alkene |
| 2 | 843, 787, 762 | C-Cl | Stretching | Halo compound |
| 3 | 1662, 1643, 1626 | C=C | Stretching | Alkene |
| 4 | 2112, 2034, 2000 | C=C=N | Stretching | Ketenimine |
| 5 | 2260, 2196, 2105 | C≡C | Stretching | Alkyne |
| 6 | 2275, 2269, 2263, 2250 | O=C=O | Stretching | Carbon dioxide |
| 7 | 3200, 2890, 2850, 2700 | O-H | Stretching | Alcohol |

**Table 2.** FTIR analysis of the functional groups of degraded byproducts of DCF sodium by the $Fe^{2+}$/PMS process.

| Sr. No | Transmittance at Wavenumber (cm$^{-1}$) | Group Species | Vibration Mode | Compound Class |
|--------|------------------------------------------|---------------|----------------|----------------|
| 1 | 724, 696, 673 | C=C | Bending | Alkene |
| 2 | 843, 787, 762 | C-Cl | Stretching | Halo compound |
| 3 | 1662, 1643, 1626 | C=C | Stretching | Alkene |
| 4 | 2112, 2034, 2000 | C=C=N | Stretching | Ketenimine |
| 5 | 2260, 2196, 2105 | C≡C | Stretching | Alkyne |
| 6 | 2230, 2243, 2256 | C≡N | Stretching | Nitrile |
| 7 | 2275, 2269, 2263, 2250 | O=C=O | Stretching | Carbon dioxide |
| 8 | 3200, 2890, 2850, 2700 | O-H | Stretching | Alcohol |
| 9 | 2987, 2913, 2845 | N-H | Stretching | Amine |

**Table 3.** FTIR analysis of the functional groups of degraded byproducts of DCF sodium using the $Fe^{2+}/H_2O_2$ process.

| Sr. No | Transmittance at Wavenumber (cm$^{-1}$) | Group Species | Vibration Mode | Compound Class |
|---|---|---|---|---|
| 1 | 730, 710, 688, 665 | C=C | Bending | Alkene |
| 2 | 810, 774, 734 | C-Cl | Stretching | Halo compound |
| 3 | 1662, 1643, 1626 | C=C | Stretching | Alkene |
| 4 | 2112, 2034, 2000 | C=C=N | Stretching | Ketenimine |
| 5 | 2275, 2269, 2263, 2250 | C=C=O | Stretching | Isothiocyanate |
| 6 | 3200, 2890, 2850, 2700 | O-H | Stretching | Alcohol |
| 7 | 3015, 3085, 2923 | N-H | Stretching | Amine |

*3.7. Mineralization Studies*

The mineralization of DCF sodium by the $Fe^2/PS$, $Fe^{2+}/PMS$ and $Fe^{2+}/H_2O_2$ processes was studied using total carbon (TC) removal, and the results are shown in Table 4. The results show that the concentration of TC decreased from 0.85 to 0.33 mg/L (i.e., 61% TC removal) in 60 min by using the $Fe^{2+}/PS$ process, while the concentration of TC was reduced from 0.85 to 0.47 mg/L (i.e., 46% TC removal) in 60 min via the $Fe^{2+}/PMS$ system. On the other hand, $Fe^{2+}/H_2O_2$ showed the lowest TC removal efficiency, represented by 33% TC removal (i.e., TC reduction from 0.85 to 0.57 mg/L) in 60 min. The results showed that the highest TC removal efficiency was exhibited by the $Fe^{2+}/PS$ system, consistent with the degradation efficiency of Diclofenac (DCF) sodium.

**Table 4.** TC removal for DCF sodium by the $Fe^{2+}/PS$, $Fe^{2+}/PMS$ and $Fe^{2+}/H_2O_2$ processes.

| Reaction Time (min) | TC (mg/L) Removal from DCF Sodium | | |
|---|---|---|---|
| | $Fe^{2+}/PS$ | $Fe^{2+}/PMS$ | $Fe^{2+}/H_2O$ |
| 0 | 0.85 | 0.85 | 0.85 |
| 10 | 0.73 | 0.78 | 0.78 |
| 20 | 0.64 | 0.72 | 0.71 |
| 30 | 0.52 | 0.67 | 0.66 |
| 40 | 0.43 | 0.62 | 0.63 |
| 50 | 0.37 | 0.54 | 0.59 |
| 60 | 0.33 | 0.47 | 0.57 |

*3.8. Implications and Limitations*

The degradation of DCF sodium by the $Fe^2/PS$, $Fe^{2+}/PMS$ and $Fe^{2+}/H_2O_2$ processes has several implications as well as limitations in practical applications. Iron (i.e., $Fe^{2+}$) is a comparatively environmentally friendly element, and traces of iron could be found in the surface and ground water resources that may take part in catalyzing the oxidation degradation of the water pollutants, such as DCF sodium. However, the high concentrations of $Fe^{2+}$ discharged into the environment from the Fenton-like processes may cause additional pollution as well. The synergistic effect of the sunlight on the efficiency of the $Fe^2/PS$, $Fe^{2+}/PMS$ and $Fe^{2+}/H_2O_2$ processes for the degradation of DCF sodium may be investigated in future studies for sustainability purposes.

**4. Conclusions**

Various homogeneous catalytic AOPs, i.e., $Fe^{2+}/PS$, $Fe^{2+}/PMS$ and $Fe^{2+}/H_2O_2$ systems, were applied for the degradation of DCF sodium in an aqueous solution. The highest degradation efficiency was shown by the $Fe^{2+}/PS$ process, represented by 89% DCF sodium

removal in 60 min. The degradation efficiency of DCF sodium was affected by the presence of different anionic species, i.e., $NO_3^-$, $HCO_3^-$ and $SO_4^{2-}$. The kinetics study showed that the degradation of DCF sodium by the studied AOPs followed pseudo-first-order kinetics. The studied AOPs resulted in significant removal of total carbon (TC) as well, represented by 61, 46 and 33% TC removal by the $Fe^{2+}$/PS, $Fe^{2+}$/PMS and $Fe^{2+}$/$H_2O_2$ systems, respectively, during 60 min. The FTIR analysis revealed that prior to mineralization, the DCF sodium was transformed into less toxic compounds, such as Alkene, alkyne, and amine, by the $Fe^{2+}$/PS, $Fe^{2+}$/PMS and $Fe^{2+}$/$H_2O_2$ systems. It was concluded that the $Fe^{2+}$/oxidant homogenous catalytic system, particularly $Fe^{2+}$/PS, was the most promising method for the elimination of toxic pharmaceuticals, i.e., DCF sodium, from the water environment.

**Author Contributions:** Investigation, W.A., N.P. and S.K.Z.; Writing—original draft preparation, W.A.; Writing—review and editing, F.R., N.P., S.K.Z., S.K. and CH; Experimentation, W.A.; Supervision, F.R.; Project administration, F.R. and C.H. All authors have read and agreed to the published version of the manuscript.

**Funding:** National Research Foundation of Korea (NRF) grant funded by the Korean government (MSIT) (No. 2021R1A2C1093183) and (No. 2021R1A4A1032746).

**Institutional Review Board Statement:** Not applicable

**Informed Consent Statement:** Not applicable

**Data Availability Statement:** Not applicable

**Acknowledgments:** CH acknowledges the support of the National Research Foundation of Korea (NRF) grant funded by the Korean government (MSIT) (No. 2021R1A2C1093183) and (No. 2021R1A4A1032746). SK acknowledges the financial support of Women University Swabi.

**Conflicts of Interest:** The authors declare no conflict of interest.

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
