# Peer review of "The Catalytic Degradation of the Inflammatory Drug Diclofenac Sodium in Water by Fe2+/Persulfate, Fe2+/Peroxymonosulfate and Fe2+/H2O2 Processes: A Comparative Analysis"

_water, doi:10.3390/w15050885_

Round 1

Reviewer 1 Report

The research article i.e. “Catalytic degradation of inflammatory drug, Diclofenac sodium in water by Fe2+/PS, Fe2+/PMS and Fe2+/H2O2 processes: Comparative analysis” by Rehman et al. focuses on degradation of diclofenac sodium by the various iron doped samples. There are certain critical issues within the article which are listed below.

1.      Abstract needs major revision. It should be free from abbreviations. In addition, it should contain the outline of the work carried out. In stead of first three line it should have only one line of the introduction of the work. Rest of the abstract should be on the work performed by the authors.

2.      There is correct in figure 1. It should be the ionic bond between sodium and oxygen. It should be revised.

3.      There are several grammatic as well as typing errors within the article that should be revised like the abbreviations should start from introduction and is first defined where is used first but the authors have defined is several times e.g. AOP etc.

4.       In Introduction, authors must include the literature of the degradation of Diclofenac sodium.

5.      In material methods, percentage purity should also be there. Also, please describe how Fe2+/PS, Fe2+/PMS have been synthesis. Is it doping or the generally the mixture of the Fe2+ and PS, Fe2+ and PMS

6.      In degradation, please provide the specification of light used.

7.      Why the authors have chosen pH 4 for the experiment. Detailed experiments on pH effect is required.

8.      Errors bars is must in all the experiments. Authors should triplicate the experiments and provide the errors in all the experiments

9.      Logical reasoning is missing in results and discussion section.

1.   Authors must provide the data of pristine Fe2+ PS, PMS in degradation.

.   My major query is currently the researcher focuses on the visible light degradation and have excellent results why the authors have chosen this system as the UV light is more toxic.

1Overall the idea is good but major revision is required in almost all the sections.

Reviewer 2 Report

This manuscript studied the efficiency of three types of advanced oxidation processes on the degradation of diclofenac (DCF) sodium. Overall, this manuscript needs significant revision in many sections to enhance the overall quality as well as readers understanding of the reported findings.

Comments:

Tittle: Remove all the abbreviations from the tittle since they are not well-known to the broader readers.

Line 49: Give full name of “DCF” as it appears first time in the introduction.

Line 54: Could you provide the value of water solubility of DCF, and how the solubility changes with the change of pH of aqueous medium.

Line 60 – 62: Authors have mentioned that DCF concentration in water is 5 ug/L, while in wastewater, it is detected in the range of 2 – 10 ug/L. The reviewer is wondering whether any international standards (USEPA or WHO) about the toxicity level of DCF.  

Line 78 – 79: Could you explain why you have considered to do the experiments using these three catalytic systems, Fe2+/persulfate (PS), Fe2+/peroxymonosulfate (PMS) and Fe2+/H2O2 processes?

Overall, the introduction lacks to explain the novelty and importance of the current study by comparing the existing published studies. Thus, before the last paragraph, authors should briefly discuss the findings of the prevailing recent studies on the research topic, then clearly highlight the key questions that were investigated in this work.

In the Materials and Methods, authors should clearly mention the DCF concentration as well as catalyst concentrations used for the experiments with justification about the selection of concentration. Also, mention other experimental conditions including the initial pH and temperature.

At present, t is not clear whether the experiments were conducted just once or in triplicates since there is no information in Materials and Methods as well as there are no error bars in all the reported figures related to degradation experiments. How to convince the readers and scientific communities about the reproducibility of the reported findings.

Line 107 – 108: The DCF degradation by Fe2+/PS, Fe2+/PMS was 89 and 82%, respectively. Are they statistically significant. It is better to do some statistical analysis prior claiming that Fe2+/PS system is better in degradation of DCF than other catalytic system.

Line 111 – 121: What is the logic for compression of DCF degradation with other compounds like triphenyl phosphate, sulfamethoxazole, trimethoprim, etc. How these compounds are relevant to the DCF. In literature, there are many studies related to DCF degradation by advanced oxidation processes. It would be more meaningful to compare your results with the published articles with DCF.

https://doi.org/10.1016/j.watres.2003.09.028

https://doi.org/10.1080/09593330.2020.1770869

https://doi.org/10.1016/j.seppur.2018.04.014

https://doi.org/10.1016/j.jece.2022.108296

Line 133: Any justification/practical relevance on tests using the DCF concentration range between 0.1, 0.5, and 1.0 mM?

Line 224: “3.6. FTIR studies of DCF sodium degradation” Could you explain more about how you isolate the degradation byproducts from the reaction mixture? Are they solid products for FT-IR analysis?

Fig 4: Why you only tested the kinetics data with the Pseudo first order equation? Isn’t worth to test the data also with Pseudo second order equation?

Fig. 7: Could you explain why you have selected only these three anions namely nitrate, carbonate and sulfate?

Add a section on the mechanisms of the three different advanced oxidation processes. Also, explain why the DCF degradation efficiency is higher in Fe2+/PS system compared to others.

You have only tested the impacts of only three anions (nitrate, carbonate and sulfate) on the DCF degradation, but in the real wastewater, how the degradation rate of DCF would be affected with the varying pH and with presence of multiple anions and cations. Authors should provide in-depth discussion in results and discussion section.

At the end of results and discussion, add a section on implications and limitations of this work.

Round 2

Reviewer 1 Report

The authors have made suffice g  hanges in the revised manuscript. I am fully satisfied with the authors reponse. I recommend the acceptance of the present article.

Author Response

Thank you very much for your time for the review. 

Reviewer 2 Report

No additional comments.

Author Response

(The authors gave the same response as above.)
